# Src-FAK Signaling Mediates Interleukin 6-Induced HCT116 Colorectal Cancer Epithelial–Mesenchymal Transition

**DOI:** 10.3390/ijms24076650

**Published:** 2023-04-02

**Authors:** Yu-Han Huang, Han-Kun Chen, Ya-Fen Hsu, Hsiu-Chen Chen, Chin-Hui Chuang, Shiu-Wen Huang, Ming-Jen Hsu

**Affiliations:** 1Division of Genetics and Genomics, Department of Pediatrics, Boston Children’s Hospital, Harvard Medical School, Boston, MA 02115, USA; 2Department of General Surgery, Chi Mei Medical Center, Tainan 710, Taiwan; 3Division of General Surgery, Department of Surgery, Landseed Hospital, Taoyuan 324, Taiwan; 4Department of Pharmacology, School of Medicine, College of Medicine, Taipei Medical University, Taipei 110, Taiwan; 5Graduate Institute of Medical Sciences, College of Medicine, Taipei Medical University, Taipei 110, Taiwan; 6Department of Medical Research, Taipei Medical University Hospital, Taipei 110, Taiwan; 7Research Center of Thoracic Medicine, Taipei Medical University Hospital, Taipei 110, Taiwan; 8Cell Physiology and Molecular Image Research Center, Wan Fang Hospital, Taipei Medical University, Taipei 110, Taiwan

**Keywords:** interleukin-6 (IL-6), colorectal cancer, epithelial–mesenchymal transition (EMT), Src, focal adhesion kinase (FAK), signal transducer and activator of transcription 3 (STAT3)

## Abstract

Colorectal cancer is one of the most prevalent and lethal malignancies, affecting approximately 900,000 individuals each year worldwide. Patients with colorectal cancer are found with elevated serum interleukin-6 (IL-6), which is associated with advanced tumor grades and is related to their poor survival outcomes. Although IL-6 is recognized as a potent inducer of colorectal cancer progression, the detail mechanisms underlying IL-6-induced colorectal cancer epithelial–mesenchymal transition (EMT), one of the major process of tumor metastasis, remain unclear. In the present study, we investigated the regulatory role of IL-6 signaling in colorectal cancer EMT using HCT116 human colorectal cancer cells. We noted that the expression of epithelial marker E-cadherin was reduced in HCT116 cells exposed to IL-6, along with the increase in a set of mesenchymal cell markers including vimentin and α-smooth muscle actin (α-SMA), as well as EMT transcription regulators—twist, snail and slug. The changes of EMT phenotype were related to the activation of Src, FAK, ERK1/2, p38 mitogen-activated protein kinase (p38MAPK), as well as transcription factors STAT3, κB and C/EBPβ. IL-6 treatment has promoted the recruitment of STAT3, κB and C/EBPβ toward the Twist promoter region. Furthermore, the Src-FAK signaling blockade resulted in the decline of IL-6 induced activation of ERK1/2, p38MAPK, κB, C/EBPβ and STAT3, as well as the decreasing mesenchymal state of HCT116 cells. These results suggested that IL-6 activates the Src-FAK-ERK/p38MAPK signaling cascade to cause the EMT of colorectal cancer cells. Pharmacological approaches targeting Src-FAK signaling may provide potential therapeutic strategies for rescuing colorectal cancer progression.

## 1. Introduction

The cancer burden continues to grow globally, according to annual statistics reported by the World Health Organization (WHO). Poor prognostic outcomes of cancer patients are closely associated with tumor metastasis, which accounts for approximately 90% of cancer-related deaths [1,2]. Angiogenesis and lymphangiogenesis have been recognized as main routes of the metastatic spread of tumor cells [3,4]. These processes are mediated by angiogenic or lymphangiogenic factors secreted by tumor or stroma cells, interacting with their cognate receptors on endothelial cells [4]. Tumor microenvironment (TME)-derived factors may also lead to metastatic plasticity via inducing epithelial–mesenchymal transition (EMT) to facilitate the dissociation of cancer cells from the primary tumor and dissemination into blood and lymphatic vessels [5,6,7,8].

Aberrant activation of EMT has been implicated in the enhanced metastasis and unfavorable prognosis in patients with colorectal cancer (CRC). EMT is a cellular process that enables polarized epithelial cells to lose their cell–cell adhesion and acquire the phenotype of mesenchymal cells with migratory and invasive properties [6,9]. During the transition, the expressions of epithelial markers such as E-cadherin are reduced, while the expression of mesenchymal markers such as vimentin and α-smooth muscle actin (α-SMA) [10,11] become elevated. A variety of signals have been shown to orchestrate this complicated process via activating a group of core EMT transcription factors [5,6,9]. These transcription factors include snail [12], slug [13], ZEB1 [14], SIP1/ZEB2 [15] and twist [16], which are prominent transcriptional repressors of E-cadherin gene. Therapeutic strategies targeting EMT may prevent tumor dissemination and improve clinical outcomes of CRC patients.

It is believed that dysregulated inflammation is closely associated with the development of various types of cancer, including CRC [17,18]. Among pro-inflammatory cytokines, interleukin-6 (IL-6) is considered as a key mediator orchestrating inflammatory processes [19]. IL-6 signaling may also facilitate tumor survival, proliferation, migration, as well as chemoresistance, which contribute to tumor progression and poor outcomes in cancer patients [20]. Growing evidence shows that IL-6 has been integral to the pathogenesis of sporadic and inflammation-associated CRC [21]. The correlation between elevated serum IL-6 levels and advanced tumor stages, increased tumor size and decreased survival of patients with CRC has also been documented [22,23,24]. IL-6 was shown to promote CRC invasion and metastasis via the induction of EMT [25]. However, the underlying mechanisms remain to be fully defined. In this study, we demonstrated that IL-6 activates the Src-FAK signaling cascade, leading to the recruitment of C/EBPβ, NFκB and STAT3 to the promoter region of Twist to increase Twist expression and induce EMT in HCT116 colorectal cancer cells.

## 2. Results

### 2.1. IL-6 Induces Epithelial–Mesenchymal Transition (EMT) in HCT116 Colorectal Cancer Cells

IL-6 binds to membrane-bound IL-6 receptor (mIL-6R; also known as IL-6Rα) and subsequently forms a complex with signaling transducing receptor glycoprotein130 (gp130; also known as IL-6Rβ) to trigger downstream classic signaling cascade. Alternatively, an IL-6 signaling complex may also be formed by the association of IL-6, soluble form of IL-6R (sIL-6R) and gp130, which initiates the trans-signaling pathway [26,27] and plays a key role in promoting tumorigenesis [27,28,29]. Therefore, we examined EMT in HCT116 human CRC cells after IL-6 or IL-6 plus sIL-6R (IL-6/sIL-6R) exposure. A 72 h treatment with IL-6 (20 ng/mL) significantly reduced E-cadherin protein levels (Figure 1A). The protein levels of E-cadherin were further reduced in the presence of IL-6 (20 ng/mL) plus sIL-6R (20 ng/mL) (IL-6/sIL-6R) (Figure 1A). IL-6/sIL-6R also led to the reduction of E-cadherin *cdh1* mRNA levels (Figure 1B), suggesting that the reduction of E-cadherin protein level is attributed to the decrease in E-cadherin *cdh1* mRNA in HCT116 cells after IL-6/sIL-6R exposure. Immunofluorescence staining for E-cadherin, which localizes at the plasma membrane, also indicates that E-cadherin levels were reduced after 72 h treatment of IL-6/sIL-6R (Figure 1C). These results are consistent with the observations reported recently that IL-6 is capable of reducing E-cadherin levels in HCT116 cells [30,31]. In addition to the epithelial marker E-cadherin, we also determined the effects of IL-6/sIL-6R on mesenchymal markers. As shown in Figure 1D, IL-6 (20 ng/mL) significantly increased protein levels of α-SMA, vimentin, snail, slug, as well as twist in HCT116 cells after exposure to IL-6 for 48 h. This increase was more pronounced in IL-6/sIL-6R-stimulated HCT116 cells (Figure 1D). Moreover, the mRNA levels of these mesenchymal markers were also increased in HCT116 cells after 48 h exposure to IL-6/sIL-6R (Figure 1E). These results suggest that IL-6 signaling is capable of inducing EMT in HCT116 cells.

### 2.2. STAT3 Mediates IL-6-Induced EMT in HCT116 Cells

Transcription factor STAT3 is the main downstream effector of IL-6 signaling [32]. We thus examined whether IL-6-induced EMT in HCT116 cells involves STAT3. As shown in Figure 2A, IL-6/sIL-6R caused an increase in STAT3 phosphorylation in a time-dependent manner (Figure 2A). Knockdown STAT3 with *stat3* siRNA significantly inhibited IL-6/sIL-6R-induced elevation of mesenchymal markers including α-SMA, vimentin, snail, slug and twist (Figure 2B). It suggests that IL-6/sIL-6R promotes EMT in HCT116 CRC cells through, at least in part, STAT3 activation. In addition to STAT3, the involvement of transcription factors such as NF-κB [33] and C/EBPβ [34] in EMT has also been described in human cancers, leading us to explore the possible association between IL-6 and NF-κB or C/EBPβ in HCT116 cells. As shown in Figure 2C, IL-6/sIL-6R treatment led to an increase in NF-κB subunit p65 phosphorylation. IL-6/sIL-6R also induced C/EBPβ phosphorylation in a time-dependent manner in HCT116 cells (Figure 2D). It raises the possibility that IL-6/sIL-6R may activate these transcription factors to upregulate EMT-transcription factors through transcriptional induction. Results derived from TFBIND (T. Tsunoda, and T. Takagi: Estimating Transcription Factor Bind ability on DNA.; http://tfbind.hgc.jp/ (accessed on 4 July 2021)) [35] analysis showed that the putative consensus sequences for STAT3, p65 and C/EBPβ are included in the 5′-flanking promoter region between −742 and −349 of the human *twist* gene. To examine whether these transcription factors are recruited to the endogenous *twist* promoter region in response to IL-6/sIL-6R, a chromatin immunoprecipitation (ChIP) analysis was performed. As shown in Figure 2E, the binding of C/EBPβ, p65 and STAT3 to the *twist* promoter region (−742/−349) was increased after IL-6/sIL-6R exposure. These observations suggest that IL-6/sIL-6R-induced C/EBPβ, p65 and STAT3 activation and subsequent twist upregulation, which may contribute to IL-6/sIL-6R-induced EMT in HCT116 CRC cells.

### 2.3. IL-6 Induces the Phosphorylation of Src, FAK, ERK1/2 and p38MAPK in HCT116 Cells

We next explored the signaling cascades involved in IL-6/sIL-6R-induced EMT in HCT116 cells. Given our previous observation that Src-FAK signaling contributes to IL-6′s actions in lymphangiogenesis induction [36], we thus examined whether Src or FAK phosphorylation is also altered in IL-6/sIL-6R-stimulated HCT116 cells. As shown in Figure 3A, IL-6/sIL-6R led to increasing Src phosphorylation in HCT116 cells, which also resulted in the incline of FAK phosphorylation in a time-dependent manner (Figure 3B). ERK1/2 and p38MAPK, known as upstream regulators of NF-κB and C/EBPβ [36,37], were also activated via IL-6/sIL-6R, as shown in Figure 3C,D, with increasing phosphorylation time dependent on the activation.

### 2.4. Src-FAK Signaling Contributes to IL-6/sIL-6R’s Effects in HCT116 Cells

To establish the causal role of Src signaling in IL-6/sIL-6′s actions in HCT116 cells, an Src inhibitor, PP2, was used. As shown in Figure 4, PP2 significantly inhibited IL-6/sIL-6R-induced phosphorylation of Src (Figure 4A), FAK (Figure 4B), ERK1/2 (Figure 4C) and p38MAPK (Figure 4D). IL-6/sIL-6R-induced phosphorylation of C/EBPβ (Figure 4E), p65 (Figure 4F) and STAT3 (Figure 4G) were also reduced in the presence of PP2. These results indicated that Src contributes to IL-6/sIL-6R-induced FAK, ERK1/2, p38MAPK, C/EBPβ, p65 and STAT3 phosphorylation in HCT116 cells. Similarly, NSC 667249, a FAK inhibitor, not only reduced IL-6/sIL-6R-induced FAK (Figure 5A), ERK1/2 (Figure 5B) and p38MAPK (Figure 5C) phosphorylation, but also significantly inhibited C/EBPβ (Figure 5D), p65 (Figure 5E) and STAT3 (Figure 5F) phosphorylation in IL-6/sIL-6R-stimulated HCT116 cells. However, NSC 667249 was without effects on IL-6/sIL-6R-induced Src phosphorylation (Figure 5G). It revealed that FAK acts as a downstream effector of Src and that Src-FAK signaling plays an essential role in IL-6/sIL-6R-activated ERK1/2, p38MAPK, C/EBPβ, p65 and STAT3 in HCT116 cells.

### 2.5. IL-6-Induced C/EBPβ and p65 Phosphorylation via ERK1/2 and p38MAPK Signaling

To ascertain the linkage between ERK1/2 or p38MAPK and transcription factors C/EBPβ, p65 and STAT3, ERK1/2 signaling inhibitor peptide I (ERK-I) and p38MAPK inhibitor III (p38-I) were employed. As shown in Figure 6A, both inhibitors significantly reduced p65 and C/EBPβ phosphorylation in HCT116 cells after IL-6/sIL-6R exposure. In contrast, IL-6/sIL-6R-induced STAT3 phosphorylation was not altered in the presence of ERK-I or p38-I (Figure 6B). These results indicated that IL-6/sIL-6R-activated NFκB and C/EBPβ could be attributed to the activation of ERK1/2 and p38MAPK signaling. However, there may be no association between ERK1/2 or p38MAPK signaling and STAT3 phosphorylation under IL-6/sIL-6R stimulation.

### 2.6. Src Signaling Contributes to IL-6/sIL-6R-Induced Epithelial–Mesenchymal Transition in HCT116 Cells

We next examined whether Src signaling contributes to the mesenchymal phenotype change induced via IL-6/sIL-6R in HCT116 cells. As shown in Figure 6C, Src signaling blockade by PP2 significantly restored E-cadherin reduction in HCT116 cells after 72 h exposure to IL-6/sIL-6R. IL-6/sIL-6R-induced elevation of twist, vimentin, snail, slug and α-SMA were also suppressed by PP2 in HCT116 cells (Figure 6D). Epithelial cells undergoing EMT may acquire motile and invasive characteristics. We thus determined the effects of IL-6/sIL-6R on HCT116 cell motility using a transwell migration assay. As the results in Figure 6E show, 72 h treatment of IL-6/sIL-6R significantly enhanced serum (10% FBS)-induced HCT116 cell migration. However, this enhancing effect was markedly reduced in the presence of PP2 (Figure 6E). These observations indicate that IL-6/sIL-6R is capable of inducing EMT via activating the Src-FAK signaling cascade in HCT116 cells.

## 3. Discussion

EMT has been recognized as a key cellular process that promotes distant metastasis, the major cause of cancer-related deaths [38]. Targeting EMT, therefore, is crucial in preventing metastasis to improve therapeutic outcomes for cancer patients [39]. It is reported that EMT could be promoted by an inflammatory tumor microenvironment (TME) that fosters the acquisition of mesenchymal features in tumor cells [32,40]. TME, known as an important prognostic marker in cancer patients [18,41], features with various pro-inflammatory cytokines including tumor necrosis factor-α (TNF-α), IL-1β and IL-6 to build a tumor-supportive microenvironment to escape immune surveillance [18,41,42]. IL-6, which was mainly investigated in this study, was previously found to be elevated in the progression of CRC, and an IL-6 signaling blockade in a colitis-associated CRC murine model revealed a decreased tumor burden [43,44]. Many lines of evidence showed that IL-6 is capable of enhancing EMT via JAK-STAT3 signaling in a variety of human cancers such as breast cancer [32,45], head and neck squamous cell carcinoma [46], pancreatic cancer [47] and CRC [48]. IL-6 was shown to promote EMT-mediated CRC invasion and metastasis via the IL-6R-STAT3-miR-34a feedback loop [25]. However, the precise mechanism responsible for IL-6-induced CRC EMT remains incompletely understood. Here, we showed that activation of Src-FAK signaling followed by STAT3, NF-κB or C/EBPβ activation and twist expression contributes to IL-6-induced CRC EMT.

There is increasing evidence indicating that IL-6 is an abundant cytokine that presents in the TME [27,28,49,50]. IL-6 leads to STAT3 activation, playing a pivotal role in tumor initiation and progression [49]. STAT3 activation induced by IL-6 is critical in colitis-associated CRC growth [44] and is known as the main mediator of EMT induction [32]. Activated STAT3 acts as a regulator of gene expression [49]; E-cadherin was reported to be downregulated via STAT3 through ZEB1 in CRC cells and enhanced cell invasion [51]. In addition, Rokavec et al. [25] demonstrated that STAT3-snail signaling contributes to IL-6′acitons in shifting CRC cells toward a mesenchymal state that is privileged to metastatic spread. As ZEB1 and snail, twist has also been recognized as an E-cadherin transcriptional repressor and cooperates synergistically with snail to induce EMT [52]. In this study, STAT3 activation is causally related to IL-6-induced twist expression and EMT in CRC cells. It is likely that E-cadherin repressors regulated via STAT3, including ZEB1, twist and snail, contribute to IL-6-induced EMT in CRC. Whether these repressors act in sequence or in synergy remains to be defined. Additional works are needed to characterize the interrelationship between ZEB1, twist, snail, as well as other E-cadherin repressors in IL-6-induced CRC EMT. 

The involvement of transcription factors NF-κB [33,53] and C/EBPβ [34] in EMT has been documented in human malignancies, while the role of C/EBPβ in regulating EMT and its underlying mechanisms remains incompletely understood. Johansson et al. [54] showed that loss of C/EBPβ alters TGF-β response from growth inhibition to EMT promotion in breast cancer. In contrast, C/EBPβ is capable of enhancing EMT through indirectly elevating matrix metalloproteinase and E-cadherin repressor slug [55,56]. Here, we noted that increased NF-κB and C/EBPβ binding to the *twist* promoter region may contribute to IL-6-induced twist expression and CRC EMT. This is consistent with previous studies, which found that ERK or p38MAPK signaling lies upstream of C/EBPβ [37,57] and ERK or p38MAPK signaling inhibitor reduced IL-6-induced C/EBPβ phosphorylation, suggesting ERK or p38MAPK signaling contributes to IL-6 actions on EMT via C/EBPβ-mediated-twist induction.

IL-6 promotes tumor progression through activating a variety of downstream signaling events including JAK-STAT, PI3K-Akt and Ras-Raf-MEK-MAPK cascades [27,28,49,50]. In addition to JAK, which interplays in between IL-6/IL-6R/gp130 complex and STAT3 signaling [58], STAT3 may also be activated by other upstream kinases such as Src [36] and FAK [59]. In agreement with these observations, we noted that the Src-FAK signaling blockade significantly reduced IL-6-induced STAT3 phosphorylation. Furthermore, inhibition of Src-FAK signaling also reduced ERK and p38MAPK phosphorylation in IL-6-stimulated HCT116 cells, while these two events were independent of STAT3 activation. The link between STAT3, NF-κB and C/EBPβ-mediated EMT remains to be established. Previous studies reported the functional or physical interactions between NF-κB and C/EBP family members [60,61]. In addition, STAT3 and C/EBPβ may have bidirectional interactions via both transcriptional and translational regulation in distant cellular functions [62,63]. This raises the possibility that STAT3 may act in synergy with C/EBPβ, and further with NF-κB, to enhance the transcription of twist under IL-6-induced Src-FAK signaling in CRC cells. Whether these transcription factors activated via Src-FAK signaling further enhance functions of core EMT transcription factors as snail and ZEB1 in CRC cells with IL-6 exposure is worth clarifying.

IL-6 signaling promotes tumor EMT, metastasis and invasiveness, which is thought to be a promising target for cancer treatment [25,32,43,44,45,46,47,48]. CRC is a heterogeneous disease and different CRC cell lines may represent distinct molecular subtypes with unique gene expression patterns [64]. Although IL-6 has been shown to induce EMT in different CRC cell lines recently [25,30,31,48], the signaling mechanisms responsible for its actions may vary and need to be fully established. We demonstrated in the present study that Src-FAK signaling plays a crucial role in IL-6-induced STAT3 activation, as well as ERK/p38MAPK-NF-κB and/or C/EBPβ cascade, resulting in Twist expression and EMT in HCT116 cells (Figure 7). Our preliminary studies also showed that IL-6 causes E-cadherin reduction and increases protein levels of twist, vimentin, snail, slug and α-SMA in HT-29 colorectal cancer cells. This phenomenon is accompanied by the increased phosphorylation of Src, FAK, ERK, p38MAPK, C/EBP, as well as STAT3 (Appendix A). A similar signaling cascade has been found to contribute to IL-6-induced EMT in different CRC cell lines, suggesting that Src-FAK signaling plays a crucial role in IL-6-induced CRC EMT. It also sheds light, at least in part, on the underlying mechanisms of EMT associated with IL-6 in CRC cells. Several in vivo studies have shown that the IL-6 knockdown could attenuate tumor metastasis in mice models [65], while IL-6 over-expression has also been validated with enhancing head and neck tumor growth and EMT change, eventually leading to tumor metastasis to the lung site [46]. Given these existing studies that IL-6 indeed alters EMT change in mice tumor models and leads to tumor metastasis, future efforts will be made to develop the co-culture system of CRC cell lines with immune cells as macrophages or stroma cells to mimic the IL-6 secreting TME system and elaborate to its consequence of CRC EMT change.

The Src-FAK signaling also participates in IL-6-induced lymphangiogenesis in lymphatic endothelial cells [36,59]. It appears that targeting IL-6 and/or its downstream Src-FAK signaling could suppress tumor progression though dampening the metastatic spread of tumor cells. Several Src and FAK inhibitors are currently undergoing clinical trials, and some have been approved by the US FDA for their use in certain cancer types [66]. Recent studies reported that tocilizumab, the humanized anti-IL-6R antibody clinically utilized for treating rheumatoid arthritis, could be effective in the treatment of cancer [67,68]. Together, these findings suggest that the therapeutic spectrum of these IL-6-signaling targeting agents may be extended to preventing tumor metastasis, although the therapeutic use of these biological agents for cancer intervention remains to be evaluated and optimized [28,50].

## 4. Materials and Methods

### 4.1. Reagents

TrypLE™, Turbofect^TM^ transfection reagent and Fetal bovine serum (FBS) were purchased from Thermo Fisher Scientific (Waltham, MA, USA). Recombinant soluble IL6Rα (sIL6Rα) and IL-6 were obtained from PeproTech (Rocky Hill, NJ, USA). PP2 and NSC 667249 were purchased from Merck (Darmstadt, Germany). Normal IgG and antibodies specific for C/EBPβ (Santa Cruz Cat sc56637) and vimentin (Santa Cruz Cat sc6260) were from Santa Cruz Biotechnology (Santa Cruz, CA, USA). Antibodies against ERK1/2 (Cell Signaling Technology Cat #4695), ERK1/2 phosphorylated at threonine 202/tyrosine 204 (T202/Y204) (Cell Signaling Technology Cat #4370), FAK (Cell Signaling Technology Cat #3285), FAK phosphorylated at tyrosine 397 (Y397) (Cell Signaling Technology Cat #3283), Src (Cell Signaling Technology Cat #2123), Src phosphorylated at tyrosine 416 (Y416) (Cell Signaling Technology Cat #21013), E-cadherin (Cell Signaling Technology Cat #3195), Snail (Cell Signaling Technology Cat #3879), C/EBPβ phosphorylated at Thr235 (T235) (Cell Signaling Technology Cat #3084), p38MAPK phosphorylated at Thr180/Tyr182 (T180/Y182) (Cell Signaling Technology Cat #9211), p38MAPK (Cell Signaling Technology Cat #9217), p65 phosphorylated at Ser536 (S536) (Cell Signaling Technology Cat #3033), p65 (Cell Signaling Technology Cat #8242), STAT3 phosphorylated at Tyr705 (Y705) (Cell Signaling Technology Cat # 9131) and STAT3 (Cell Signaling Technology Cat #4904) were from Cell Signaling (Danvers, MA, USA). Antibodies against GAPDH (GeneTex Cat GTX100118), α-tubulin (GeneTex Cat #GTX628802), slug (GeneTex Cat GTX128796), Twist (GeneTex Cat GTX127310), as well as horseradish peroxidase conjugated anti-rabbit and anti-mouse IgG antibodies were from GeneTex Inc (Irvine, CA, USA). All materials for Western blot were obtained from Bio-Rad (Hercules, CA, USA). McCoy’s 5A medium, negative siRNA, human STAT3 siRNA, ERK activation inhibitor peptide I (ERK-I), p38 MAPK inhibitor III (p38-I) and all other chemicals were obtained from Sigma-Aldrich (St. Louis, MO, USA). PP2, NSC 667249 (FAK inhibitor, FAK-I), ERK-I and p38-I were dissolved in dimethyl sulfoxide (DMSO). The vehicle used in the control group in the absence of these inhibitors was 0.1% DMSO. PP2 (1 μM), NSC 667249 (1 μM), ERK-I (3 μM), p38-I (3 μM) and DMSO (0.1%) were without effects on HCT116 cell viability as determined by MTT assay (Appendix A).

### 4.2. Cell Culture

Dr. Bert Vogelstein [69] kindly provided the HCT116 cell line. HCT116 cells were maintained in McCoy’s 5A medium containing 10% FBS, 0.25 μg/mL amphotericin B, 100 μg/mL streptomycin and 100 U/mL of penicillin G (Biological Industries, Cromwell, CT, USA) in a 37 °C humidified incubator with 5% CO_2_.

### 4.3. Immunofluorescence Staining

HCT116 cells seeded on glass cover slips were stimulated with IL-6/sIL-6R for 72 h. After being washed twice with PBS, cells were fixed for 15 min at room temperature in paraformaldehyde (4% in PBS). Cells were then incubated with 0.1% Triton X-100 in PBS for 1 min for permeabilization. After being washed twice, cells were treated with 1% BSA in PBS for another 1 h. Cells were reacted with rabbit anti-E-cadherin antibody (Cell Signaling Technology, Danvers, MA, USA) (1:100 dilution in PBS) for 2 h at room temperature. Slides were washed twice and incubated with FITC-conjugated goat anti-rabbit IgG for another 1 h. DAPI containing mounting solution (SlowFad Gold, Thermo Fisher Scientific, Waltham, MA, USA) was used to mount the slides. E-cadherin distribution was then observed under a confocal microscope (Zeiss, LSM 410). Blue fluorescence (derived from DAPI) represented nuclei and green fluorescence indicated E-cadherin.

### 4.4. Western Blot

Western blot was performed as previously described [70]. Cells were lysed in an extraction buffer containing 0.5% NP-40, 2 mM phenylmethanesulfonylfluooride (PMSF), 140 mM NaCl, 10 mM Tris (pH 7.0), 0.2 mM leupeptin and 0.05 mM pepstatin A. Equal amounts of protein samples were subjected to SDS-PAGE and transferred onto a nitrocellulose membrane (Pall Corporation, Washington, NY, USA). After 1 h blocking with 5% non-fat milk-containing TBST buffer, proteins were recognized by incubation for 2 h with specific primary antibodies, followed by incubation with secondary antibodies conjugated with horse radish peroxidase for another 1 h. An enhanced chemiluminescence detection kit (Millipore, Billerica, MA, USA) was used to detect immunoreactivity as per the manufacturer’s instructions. Quantitative data were obtained using a computing densitometer with a scientific imaging system (Biospectrum AC System, UVP, Upland, CA, USA).

### 4.5. Suppression of stat3 Expression

Suppression of stat3 expression was performed as previously described [71]. Negative control siRNA and pre-designed siRNA targeting the human *stat3* were obtained from Sigma-Aldrich (St. Louis, MO, USA). The siRNA oligonucleotides were as follows: negative control siRNA, 5′-gaucauacgugcgaucaga-3′; and *stat3* siRNA, 5′-ggauaacgucauuagcaga-3′.

### 4.6. Cell Transfection

Cells (7 × 10^4^ cells/well) were transfected with negative control siRNA or human *stat3* siRNA as described above for immunoblotting using a Turbofect^TM^ transfection reagent (Thermo Fisher Scientific, Waltham, MA, USA) as per the manufacturer’s instructions.

### 4.7. Reverse-Transcription Polymerase Chain Reaction (RT-PCR)

A TRIzol^TM^ reagent (Thermo Fisher Scientific, Waltham, MA, USA) was used to isolate total RNA from HCT116 cells. A GoScript™ Reverse Transcription System (Promega, Madison, WI, USA) was employed for complementary DNA (cDNA) synthesis as per the instructions of the manufacturer. PCR was then performed with the GoTaq PCR Master Mix (Promega, Madison, WI, USA) and cycling conditions were: 2 min hot-start activation at 95 °C, followed by 30 cycles of 30 s denaturation at 94 °C, 30 s annealing at 56 °C and 45 s extension at 72 °C, respectively. Primer pairs for the transcripts were: E-cadherin *cdh1* sense, 5′-ccttagaggtgggtgactacaa-3′; E-cadherin *cdh1* anti-sense, 5′-tcagactagcagcttcggaac-3′; *twist* sense, 5′-agatgtcattgtttccagagaagg -3′; *twist* anti-sense, 5′-ctatcagaatgcagaggtgtgag-3′. *vimentin* sense, 5′-tggcacgtcttgaccttgaa-3′; *vimentin* anti-sense, 5′-ggtcatcgtgatgctgagaa-3′; *snail* sense, 5′-gctccttcgtccttctcctcta-3′; *snail* anti-sense, 5′-ggcactggtacttcttgaca-3′. *slug* sense, 5′-ctggtcaagaagcatttcaacg-3′; *slug* anti-sense, 5′-ggtaatgtgtgggtccgaata-3′; *gapdh* sense, 5′-agggctgcttttaactctggt-3′; *gapdh* anti-sense, 5′-ccccacttgattttggaggga-3′.

### 4.8. Cell Migration Assay

An 8 μm-pore-size transwell plate (Corning, NY, USA) was employed to perform the cell migration assay as described previously [70]. HCT116 cells were stimulated with IL-6/sIL-6R with or without PP2 for 72 h. After treatment, cells (2 × 10^4^ cells per well) in 200 µL serum-free McCoy’s 5A medium were seeded in the top transwell chambers. The bottom transwell chambers were filled with McCoy’s 5A medium containing 10% FBS. After 24 h incubation in a 37 °C incubator with 5% CO_2_, a cotton swab was used to scrap the non-migrated cells on the top side of the insert membrane. Four percent paraformaldehyde was used to fix the cells that had migrated through the membrane. Migrated cells were then stained with 0.5% toluidine blue O. Stained migrated HCT116 cells were observed under an inverted contrast phase light microscope (×40, Nikon, Tokyo, Japan). The migrated cells were also quantified by dissolving the stained cells using DMSO and measuring the absorbance at 590 nm. 

### 4.9. Chromatin Immunoprecipitation (ChIP) Analysis

A ChIP analysis was performed as previously described [71]. After different treatments, HCT116 cells were cross-linked at 37 °C for 10 min with 1% formaldehyde, followed by a rinse with ice-cold PBS. Cells harvested in SDS lysis buffer were sonicated four times for 15 s each. After 10 min of centrifugation, the collected supernatants were diluted in ChIP dilution buffer. An aliquot of each sample was used as “Input” in the PCR analysis. Normal IgG, STAT3, C/EBPβ or p65 antibodies were added to the soluble chromatin at 4 °C for 16 h for immunoprecipitation. Protein A-Magnetic Beads (Millipore, Billerica, MA, USA) were used to collect the immune complexes at 4 °C for another 2 h of incubation with gentle rotation. The samples were sequentially washed in the following buffers: low salt, high-salt and LiCl immune complex washing buffer. After being washed two times with Tris-EDTA buffer, the complexes were eluted two times for 100 µL aliquots of elution buffer each. After 4 h of incubation with 0.2 M, NaCl at 65 °C was performed to reverse the cross-linked chromatin complex, GP^TM^ DNA purification spin columns (Viogene, New Taipei City, Taiwan) were employed to purify DNA. PCR was then performed with PCR Master Mix (Promega, Madison, WI, USA) as per the manufacturer’s instructions. In total, 10% of the total purified DNA was used for PCR in a 50-μL reaction mixture. The following primer pair was used to amplify the 393-bp human *twist* promoter (−349 to −742) fragment, sense: 5′-aga agc tgt tgc cat tgc tg-3′ and antisense: 5′-ctc cgt gca ggc gga aag ttt gg-3′. The cycling conditions were: 95 °C denaturation for 5 min, followed by 30 cycles of 30 s denaturation at 94 °C, 30 s annealing at 56 °C and 45 s extension at 72 °C, respectively. After final extension for another 10 min at 72 °C, agarose gel (1.5%) electrophoresis was performed to analyze the PCR products.

### 4.10. Data Analysis

Data and statistical analyses were performed as previously described [71]. Briefly, results are represented as mean ± standard error of mean (SEM) (*n* ≥ 3). Normalization was performed to compare the differences after the treatment to control for unwanted sources of variation and to reveal relevant trends. The protein expression levels and the status of protein modification in IL-6/sIL-6R-stimulated cells were expressed as fold changes over those of the control group, whose expression was set to 1. SigmaPlot 14 (Build 10.0.0.54; Systat Software, San Jose, CA, USA; SigmaPlot) was used to perform statistical analyses. A *p* value smaller than 0.05 was defined as statistically significant.

## Figures and Tables

**Figure 1 ijms-24-06650-f001:**
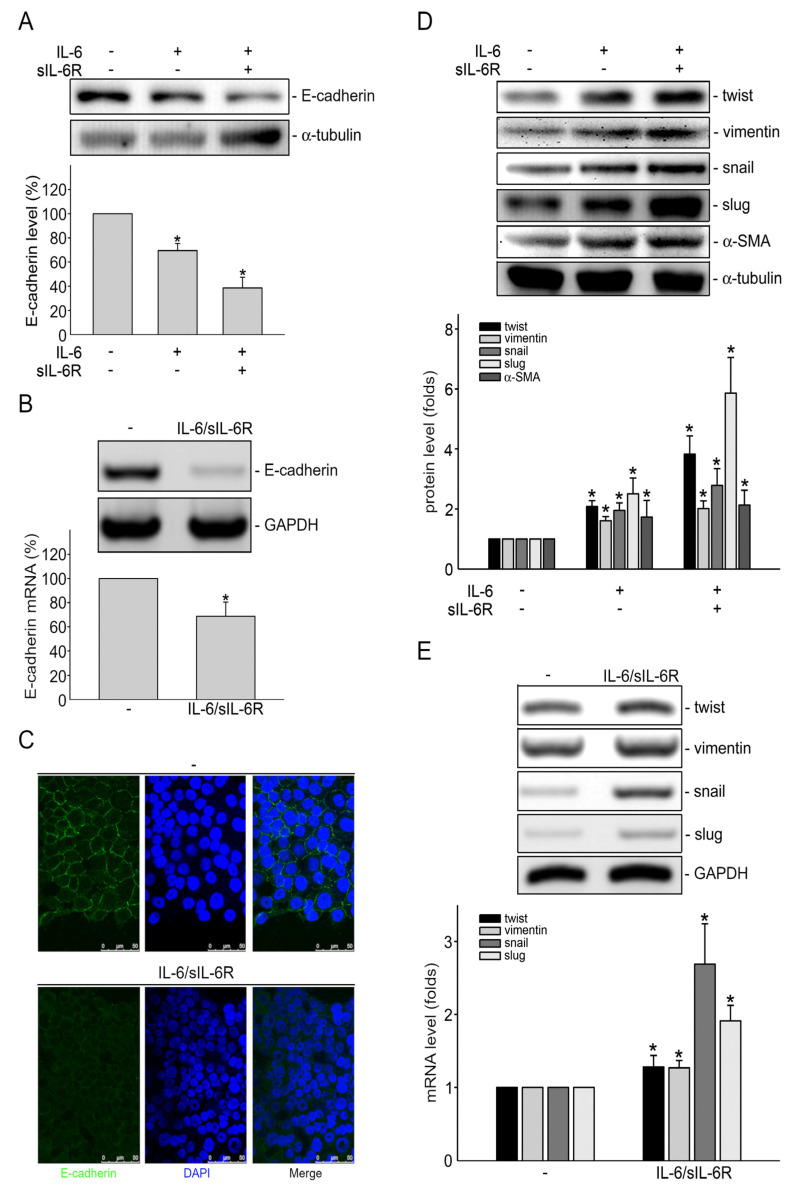
IL-6 induces epithelial–mesenchymal transition (EMT) in HCT116 colorectal cancer cells. (**A**) After 72 h treatment of vehicle with IL-6 (20 ng/mL) or IL-6 plus sIL6R (20 ng/mL), the extent of E-cadherin protein level was determined by immunoblotting. Bar graph represents the mean ± S.E.M. of six independent experiments. * *p* < 0.05, compared with the control group. (**B**) After 48 h stimulation of vehicle with IL-6 (20 ng/mL) or IL-6/sIL6R (20 ng/mL), HCT116 cells were harvested and RT-PCR was employed to examine E-cadherin *cdh1 mRNA* level as described in the “Materials and Methods” Section 4. Bar graph represents the mean ± S.E.M. of five independent experiments. * *p* < 0.05, compared with control group. (**C**) After treatment as described in (**A**), immunofluorescence analysis was employed to examine E-cadherin distribution as described in “Materials and Methods” Section 4. Results shown are representative of three independent experiments. (**D**) After treatment as described in (**B**). Immunoblotting was used to determine the protein levels of Twist, vimentin, snail, slug and α-SMA. Bar graph represents the mean ± S.E.M. of three independent experiments. * *p* < 0.05, compared with the control group. (**E**) After 24 h treatment of vehicle with IL-6 (20 ng/mL) or IL-6/sIL6R (20 ng/mL). RT-PCR was used to examine the *mRNA* level of *twist*, *vimentin*, *snail* and *slug* as described in Section 4. Bar graph represents the mean ± S.E.M. of three independent experiments. * *p* < 0.05, compared with the control group. IL-6/sIL6R: IL-6 plus sIL6R.

**Figure 2 ijms-24-06650-f002:**
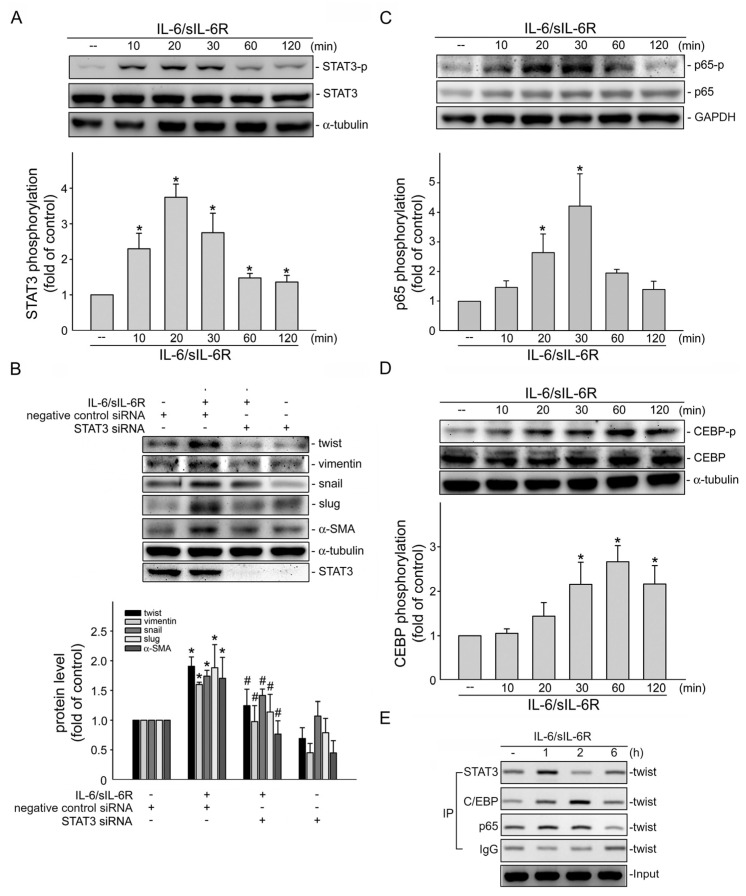
STAT3 mediates IL-6/sIL6R-induced EMT in HCT116 cells. (**A**) HCT116 cells were stimulated by IL-6/sIL-6R (20 ng/mL) for indicated time periods. After stimulation, immunoblotting was employed to determine the STAT3 phosphorylation status. Bar graph represents the mean ± S.E.M. of five independent experiments. * *p* < 0.05, compared with the control group. (**B**) HCT116 cells were transfected transiently with STAT3 siRNA or negative control siRNA for 48 h. After transfection, cells were stimulated by IL-6/sIL-6R (20 ng/mL) for another 48 h. Immunoblotting was used to examine the protein levels of Twist, vimentin, Snail, slug, α-SMA or STAT3. Bar graph represents the mean ± S.E.M. of four independent experiments. * *p* < 0.05, compared with the control group; ^#^ *p* < 0.05, compared with the group treated with IL-6/sIL6R (20 ng/mL). HCT116 cells were stimulated by IL-6/sIL6R (20 ng/mL) for different time periods as indicated. The p65 (**C**) or C/EBP (**D**) phosphorylation status was examined using immunoblotting. Bar graph represents the mean ± S.E.M. of five independent experiments. * *p* < 0.05, compared with the control group. (**E**) Cells were stimulated by IL-6/sIL-6R (20 ng/mL) for 1, 2 or 6 h. A ChIP assay was performed as described in “Materials and Methods” Section 4. Typical traces representative of three independent experiments with similar results are shown. Before immunoprecipitation, the promoter region (−742/−349) of Twist was detected in the cross-linked chromatin sample (input, positive control).

**Figure 3 ijms-24-06650-f003:**
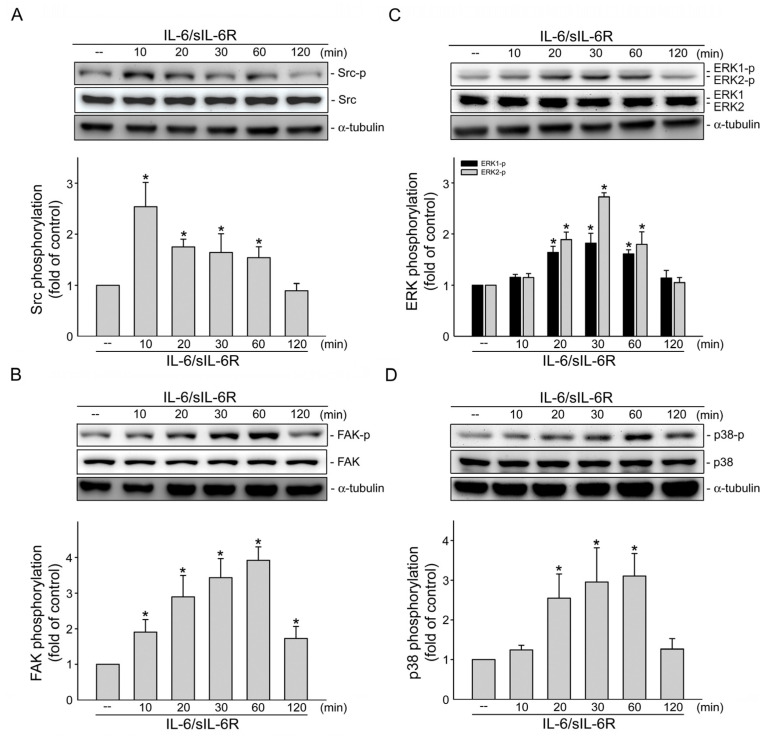
Src, FAK, ERK1/2 and p38MAPK phosphorylation were increased in IL-6/sIL-6R-stimulated HCT116 cells. HCT116 cells were stimulated by IL-6/sIL-6R (20 ng/mL) for different time periods as indicated. Immunoblotting was employed to examine the Src (**A**), FAK (**B**), ERK1/2 (**C**) and p38MAPK (**D**) phosphorylation status. Bar graph represents the mean ± S.E.M. of five independent experiments. * *p* < 0.05, compared with the control group.

**Figure 4 ijms-24-06650-f004:**
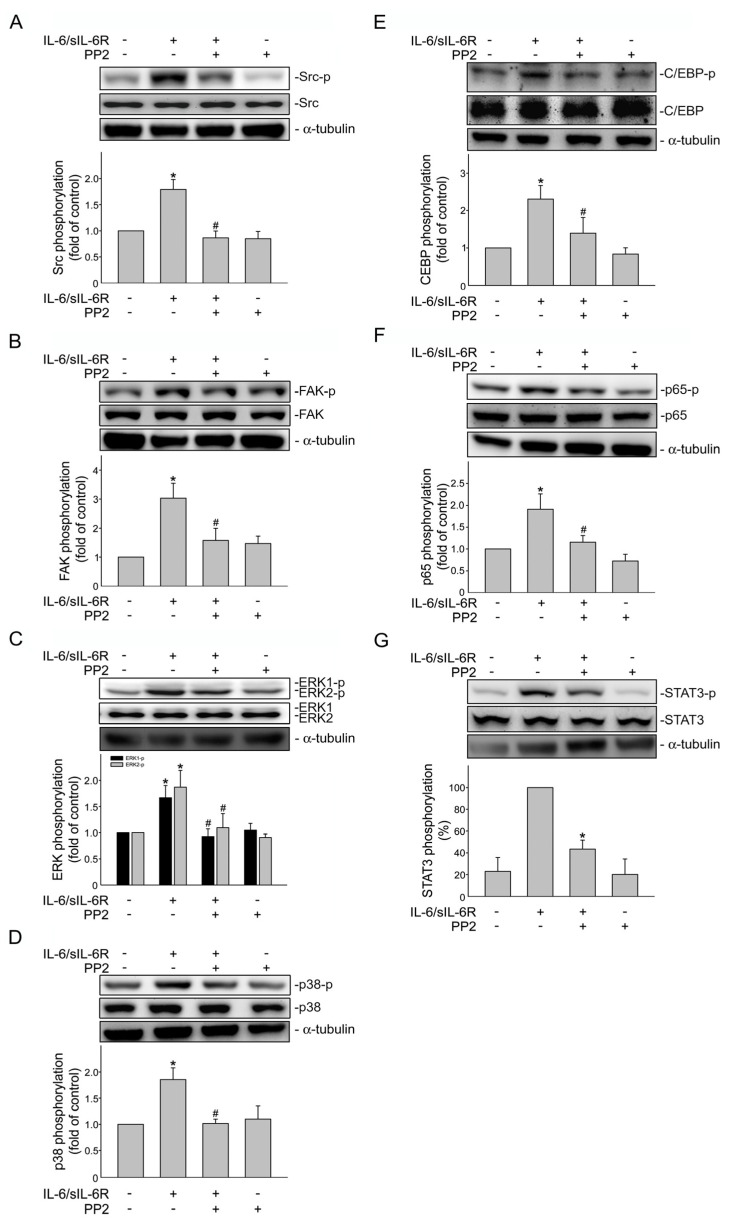
Src mediates IL-6-induced FAK, ERK1/2, p38MAPK, p65, C/EBP and STAT3 phosphorylation in HCT116 cells. Cells were treated with PP2 (1 μM) for 30 min and stimulated by IL-6/sIL6R (20 ng/mL) for another 30 min. Immunoblotting was used to examine Src (**A**), FAK (**B**), ERK1/2 (**C**), p38MAPK (**D**), p65 (**E**), C/EBP (**F**) and STAT3 (**G**) phosphorylation status. Bar graph represents the mean ± S.E.M. of four independent experiments. * *p* < 0.05, compared with the control group. ^#^
*p* < 0.05, compared with the group treated with IL-6/sIL-6R alone.

**Figure 5 ijms-24-06650-f005:**
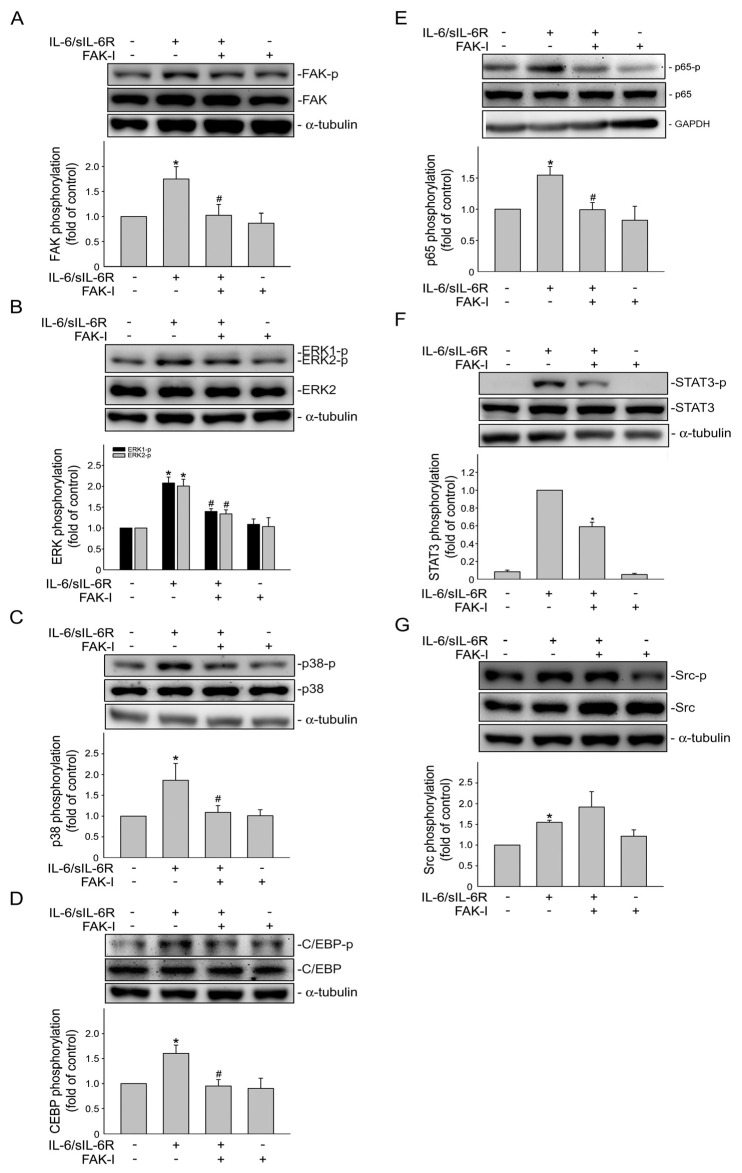
FAK participates in ERK1/2, p38MAPK, C/EBP, p65 and STAT3 phosphorylation in IL-6/sIL-6R-stimulated HCT116 cells. Cells were treated with FAK inhibitor (FAK-I) (1 μM) for 30 min, followed by stimulation with IL-6/sIL6R (20 ng/mL) for another 30 min. Immunoblotting was used to examine FAK (**A**), ERK1/2 (**B**), p38MAPK (**C**), C/EBP (**D**), p65 (**E**), STAT3 (**F**) and Src (**G**) phosphorylation status. Bar graph represents the mean ± S.E.M. of four independent experiments. * *p* < 0.05, compared with the control group. ^#^
*p* < 0.05, compared with the group treated with IL-6/sIL-6R alone.

**Figure 6 ijms-24-06650-f006:**
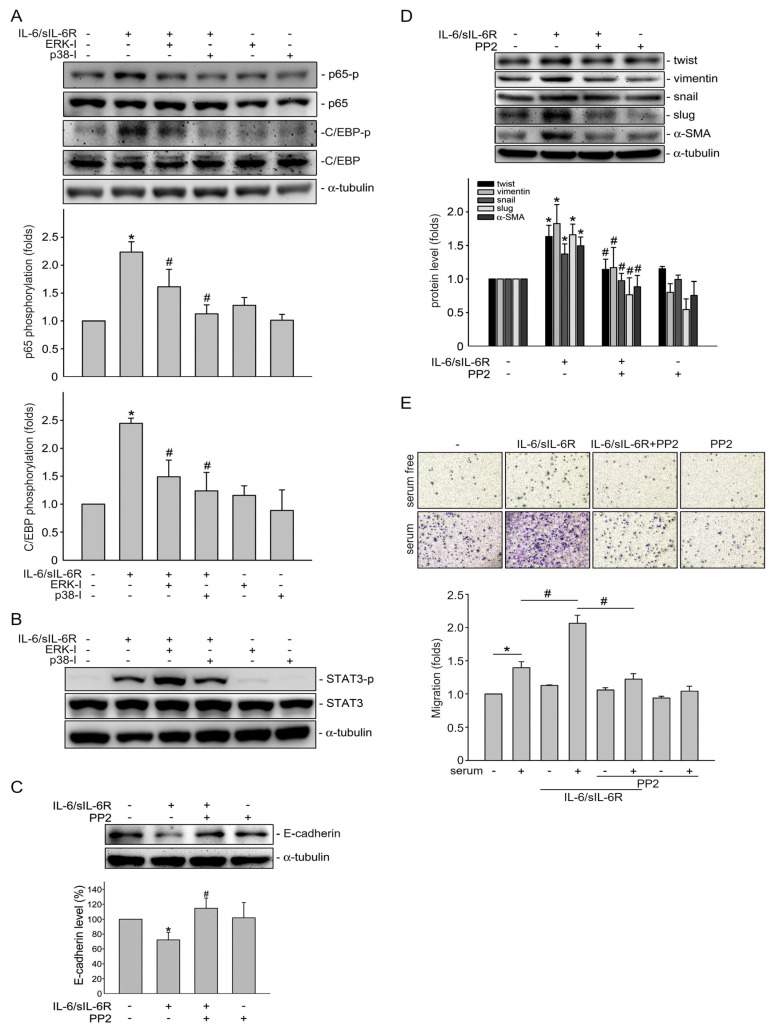
Inhibiting Src signaling attenuated IL-6/sIL-6R-induced EMT in HCT116 cells. HCT116 cells were treated with ERK inhibitor (ERK-I) (3 μM) or p38MAPK inhibitor (p38-I) (3 μM) for 30 min, followed by stimulation with IL-6/sIL6R (20 ng/mL) for another 30 min. Immunoblotting was used to examine the p65 or C/EBP phosphorylation status. Bar graph represents the mean ± S.E.M. of four independent experiments. * *p* < 0.05, compared with control group. (**B**) After treatment as described in (**A**), STAT3 phosphorylation status was examined using immunoblotting. Typical traces representative of three independent experiments with similar results are shown. HCT116 cells were treated for 30 min with 1 μM PP2 and stimulated by IL-6/sIL-6R (20 ng/mL) for another 72 (**C**) or 48 h (**D**). Immunoblotting was used to examine the protein levels of E-cadherin (**C**), twist, vimentin, snail, slug and α-SMA. Bar graph represents the mean ± S.E.M. of five independent experiments. **p* < 0.05, compared with the control group. (**E**) After 30 min treatment with vehicle or 1 μM PP2, cells were stimulated with IL-6/sIL6R (20 ng/mL) for another 72 h. Cells were resuspended in serum-free McCoy’s 5A medium and seeded in the top chamber of the transwell plate. Cells were then allowed to migrate for 24 h toward serum (10% FBS). Stained HCT116 cells that had migrated through the membrane were quantified as described in the “Materials and Methods” Section 4. Bar graph represents the mean ± S.E.M. of five independent experiments. **p* < 0.05, compared with the serum-free control group. ^#^ *p* < 0.05, compared with the group primed with IL-6/sIL6R alone.

**Figure 7 ijms-24-06650-f007:**
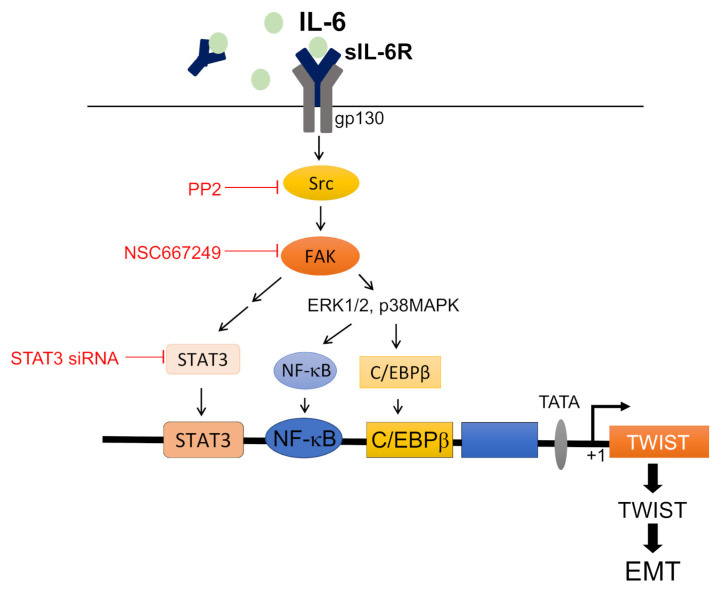
Schematic summary of the signaling cascade involved in IL-6-induced Twist expression and EMT in HCT116 colorectal cancer cells.

## Data Availability

Not applicable.

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
