# Peer review of "Src-FAK Signaling Mediates Interleukin 6-Induced HCT116 Colorectal Cancer Epithelial–Mesenchymal Transition"

_ijms, 2023, doi:10.3390/ijms24076650_

Round 1
Reviewer 1 Report
In this interesting paper authors introduce IL-6/sIL6-R as an important pathway controlling EMT in colorectal cancer. In my opinion topic is well research and have enough data to support research hypothesis. There are few thinks I would like to see to be added to make this paper stronger and clearer in manner of presented data.
I would like to see improvement in discussion. In most of the time it is reading as a very well describe results. Can authors elaborate more on concept what their research brings novel and important to colorectal cancer research and even to oncology research.
Acceptable to publish with major revisions.
· Provide immunoblot with housekeeping proteins for Western blots
in Fig. -2,3,4,5,6
· Provide cytotoxicity assay for all inhibitors used in the study.
Include solvent controls for inhibitors.
· Provide information about ERK and p 38 inhibitor in M@M
· Provide solvent control as DMSO, used for inhibitors –for western blots.
· In Figure 1. You are using IL-6 treatment and combination of IL-6/sIL-6R it is proper to have sIL-6R as a separate treatment too. Please add them to the Figure1. Western blots data.
· Provide proper pictures of IF staining for E –Catherin, each separate staining and marched staining image. Include IgG controls for this staining. Each picture should have measuring bar.
· Migration assay is run in different condition than any other treatments in your research, what incudes extra 24 hr time in serum free media. Could you provide data confirming that alternation in main signaling is still happening for 24hr after treatments in absents of IL6 and inhibitors?
· Provide figure with graphic proposal of signaling in HCT116.
Author Response
We thank Reviewer 1’s insightful suggestions and valid critiques. Our responses are as follows:
Critique#1: Provide immunoblot with housekeeping proteins for Western blots in Fig. -2,3,4,5,6
Response: We appreciate this thoughtful and pertinent critique. In addition to the target proteins, we also performed immunoblotting with housekeeping proteins such as GAPDH or α-tubulin for each western blot experiment. In the revised manuscript, we have included these immunoblots for housekeeping proteins in Fig. 2A, 2C, 2D, Fig. 3, Fig. 4, Fig. 5, Fig. 6A, and 6B.
Critique #2: Provide cytotoxicity assay for all inhibitors used in the study. Include solvent controls for inhibitors.
Response: We appreciate this thoughtful critique. In the revised manuscript, we have conducted another set of experiments showing the effects of inhibitors including PP2, NSC 667249 (FAK inhibitor, FAK-I), ERK activation inhibitor peptide I (ERK-I), p38 MAPK inhibitor III (p38-I) on cell viability. Results from the MTT assay demonstrated that these inhibitors at 1 or 3 μM were without effects on HCT116 cell viability. In addition, these inhibitors are dissolved in dimethyl sulfoxide (DMSO). The vehicle used in the control group in the absence of these inhibitors is 0.1 % DMSO. Results from the MTT assay also showed that treatment of HCT116 cells with 0.1 % DMSO did not alter cell viability. These additional experiments and results have been added to the “Materials and Methods” (page 13, lines 362-364) section in the revised manuscript and “Supplement Information”. They are also presented in Supplement Figure S1.
Critique #3: Provide information about ERK and p 38 inhibitor in M@M
Response: We appreciate this valid critique. ERK activation inhibitor peptide I (ERK-I) and p38 MAPK inhibitor III (p38-I) were purchased from Sigma-Aldrich (St. Louis, MO, U.S.A.). This information has been added to the” Materials and methods” (page 12, lines 355-356) section in the revised manuscript.
Critique #4: Provide solvent control as DMSO, used for inhibitors –for western blots.
Response: We appreciate this valid critique. Inhibitors such as PP2, NSC 667249 (FAK inhibitor, FAK-I), ERK activation inhibitor peptide I (ERK-I), and p38 MAPK inhibitor III (p38-I) used in this study are dissolved in dimethyl sulfoxide (DMSO). The vehicle used in the control group in the absence of these inhibitors is 0.1 % DMSO. In the revised manuscript, we have added the following sentences to the “Materials and methods” (page 10, lines 11-12) section to address this issue.
” PP2, NSC 667249, ERK activation inhibitor peptide I (ERK-I) and p38 MAPK inhibitor III (p38-I) are dissolved in dimethyl sulfoxide (DMSO). The vehicle used in the control group in the absence of these inhibitors is 0.1 % DMSO.”
Critique #5: In Figure 1. You are using IL-6 treatment and combination of IL-6/sIL-6R it is proper to have sIL-6R as a separate treatment too. Please add them to the Figure1. Western blots data
Response: We appreciate this thoughtful critique. IL-6R and gp130 are central to the initiation of IL-6-dependent signaling by which IL-6 binds to membrane-bound IL-6R (mIL-6R) or soluble IL-6R (sIL-6R) and then forms the complex with gp130 subsequently. Cell treated with IL-6 only would trigger cis-signaling via IL-6/mIL-6R/gp130 complex formation, while treatment of IL-6 together with sIL-6R would enhance its effects due to additional IL-6/sIL-6R/gp130 complex formation and activation of trans-signaling. Given the fact that both IL-6 and IL-6R are required to trigger the signaling transduction, there would be no downstream effect by treating cells with sIL-6R merely due to lack of ligand, IL6
Critique #6: Provide proper pictures of IF staining for E-Catherin, each separate staining and marched staining image. Include IgG controls for this staining. Each picture should have measuring bar.
Response: We appreciate this valid critique. In the revised manuscript, we have added the separated E-Catherin or DAPI staining images, as well as the merged ones in Fig. 1C. The scale bars are also included in each image.
Critique #7: Migration assay is run in different condition than any other treatments in your research, what incudes extra 24 hr time in serum free media. Could you provide data confirming that alternation in main signaling is still happening for 24hr after treatments in absents of IL6 and inhibitors?
Response: We appreciate this valid critique. The purpose of the migration assay is to confirm whether the mesenchymal phenotype change directs to its characterized function as increasing motility. While the IL-6/sIL-6R treatment has been evaluated with increasing mesenchymal markers after 72 hours as shown in Figure 1, cells pretreated with IL-6/sIL-6R for 72 hours prior to the migration assay would be probably the mesenchymal type. Increasing cell motility as demonstrated in the migration assay suggests their eventual mesenchymal characteristics. The addition of IL-6/sIL-6R during cell migration could be meant by the further association between IL-6/sIL-6R and mesenchymal cells, which differs from our purpose in terms of the motility assay
Critique #8: Provide figure with graphic proposal of signaling in HCT116
Response: We appreciate this valid critique. In the revised manuscript, we have included the schematic summary of the signaling cascade involved in IL-6-induced Twist expression and EMT in HCT116 colorectal cancer cells. They are also presented in Fig. 7.

Reviewer 2 Report
This study is about how the authors identified and validated a signaling pathway involved in colorectal cancer epithelial-mesenchymal transition mediated by IL-6. The manuscript is well-written and the data is nicely presented. Given that the findings of this study are important for clinical translation, I have some questions and comments:
1. The first part of the manuscript describes the effects of IL-6 treatment on E-Cadherin expression in HCT116 cells. However, this has been described and shown before in other publications. Maybe can consider citing those papers.
2. One limitation of this study is that the authors only used a single cell line which also only represents one molecular subtype of CRC to demonstrate how IL-6 modulates the EMT. It would be beneficial if the authors can repeat some of the experiments using other CRC lines.
3. Another limitation of this study is that only in vitro experiments were performed. As discussed in the Introduction and Discussion, the tumor microenvironment plays an indispensable role in facilitating EMT. It would be beneficial if the authors can demonstrate the role of IL-6 in the signaling using some models (tumor organoids or mouse models) that can recapitulate the TME of colorectal cancer.
Author Response
We thank reviewer2’s insightful suggestions and valid critiques. Our responses are as follows:
REVIEWER 2
This study is about how the authors identified and validated a signaling pathway involved in colorectal cancer epithelial-mesenchymal transition mediated by IL-6. The manuscript is well-written and the data is nicely presented. Given that the findings of this study are important for clinical translation, I have some questions and comments:
Critique 1: The first part of the manuscript describes the effects of IL-6 treatment on E-Cadherin expression in HCT116 cells. However, this has been described and shown before in other publications. Maybe can consider citing those papers.
Response: We appreciate this valid critique. We have cited recent studies showing IL-6’s reducing effects on E-cadherin level in HCT116 cells and added the following sentences to the “Results” (page 2, lines 96-97) section to address this issue.
“These results are consistent with the observations reported recently that IL-6 is capable of reducing E-cadherin levels in HCT116 cells [1,2].”
Critique 2: One limitation of this study is that the authors only used a single cell line which also only represents one molecular subtype of CRC to demonstrate how IL-6 modulates the EMT. It would be beneficial if the authors can repeat some of the experiments using other CRC lines.
Response: We appreciate this thoughtful critique. In the revised manuscript, we have conducted another set of experiments showing the effects of IL-6/sIL-6R on epithelial and mesenchymal markers in HT-29 colorectal cancer cells. Results derived form immunoblotting showed that IL-6/sIL-6R causes E-cadherin reduction and increases protein levels of twist, vimentin, snail, slug and α-SMA in HT-29 cells. This phenomenon is accompanied by the increased phosphorylation of Src, FAK, ERK, p38MAPK, as well as C/EBP as determined by immunoblotting. Similar signaling cascade has been found to contribute to IL-6-induced EMT in different CRC cell lines, suggesting that Src-FAK signaling plays a crucial role in IL-6-induced CRC EMT. These additional experiments and results have been added to the “Supplement Information”. They are also presented in Supplement Figure S2. We also added the following sentences to the “Discussion” (page 12, lines 318-329) section to address this issue.
“CRC is a heterogeneous disease and different CRC cell lines may represent distinct molecular subtypes with unique gene expression patterns [3]. Although IL-6 has been shown to induce EMT in different CRC cell lines recently [1,2,4,5], the signaling mechanisms responsible for its actions may vary and needs to be fully established. In this study, we demonstrated that Src-FAK signaling plays a causal role in IL-6-induced STAT3 activation as well as ERK/p38MAPK-NF-κB and/or C/EBPβ cascade, leading to Twist expression and EMT in HCT116 cells (Figure 7). Our preliminary studies also showed that IL-6 causes E-cadherin reduction and increases protein levels of twist, vimentin, snail, slug and α-SMA in HT-29 colorectal cancer cells. This phenomenon is accompanied by the increased phosphorylation of Src, FAK, ERK, p38MAPK, STAT3, as well as C/EBP (Supplement Figure S2). Similar signaling cascade has been found to contribute to IL-6-induced EMT in different CRC cell lines, suggesting that Src-FAK signaling plays a crucial role in IL-6-induced CRC EMT.”
Critique 3: Another limitation of this study is that only in vitro experiments were performed. As discussed in the Introduction and Discussion, the tumor microenvironment plays an indispensable role in facilitating EMT. It would be beneficial if the authors can demonstrate the role of IL-6 in the signaling using some models (tumor organoids or mouse models) that can recapitulate the TME of colorectal cancer.
Response: We appreciate this thoughtful critique. The aim of our study is to evaluate the role of IL-6 in CRC EMT progression as well as its upstream signaling cascade, and in this article, we have shown that IL-6 induced EMT by Src-FAK-ERK/p38MAPK signaling cascade in CRC cells. Concurring with the reviewer's suggestion, further in vivo investigation of the role of IL-6 derived from TME would be a solid support to our finding. Several in vivo studies have shown that the IL-6 knockdown could attenuate tumor metastasis in mice model [6], while IL-6 over-expression has also been validated with enhancing tumor growth and EMT change, eventually leading to tumor metastasis to the lung site [7]. Given these existing studies that IL-6 indeed alters EMT change in mice tumor models and leads to tumor metastasis, future efforts will be made to develop the co-culture system of CRC cell lines with immune cells as macrophages or stroma cells to mimic the IL6 secreting TME system and elaborate to its consequence of CRC EMT change. We have added several sentences to the “Discussion” (page 12, lines 332-338) section to address this issue.
References:
- Bai J, Zhang X, Shi D, Xiang Z, Wang S, Yang C, Liu Q, Huang S, Fang Y, Zhang W, Song J, Xiong B. Exosomal miR-128-3p Promotes Epithelial-to-Mesenchymal Transition in Colorectal Cancer Cells by Targeting FOXO4 via TGF-beta/SMAD and JAK/STAT3 Signaling. Front Cell Dev Biol. 2021; 9: 568738.
- Kang S, Kim BR, Kang MH, Kim DY, Lee DH, Oh SC, Min BW, Um JW. Anti-metastatic effect of metformin via repression of interleukin 6-induced epithelial-mesenchymal transition in human colon cancer cells. PLoS One. 2018; 13: e0205449.
- Linnekamp JF, Hooff SRV, Prasetyanti PR, Kandimalla R, Buikhuisen JY, Fessler E, Ramesh P, Lee K, Bochove GGW, de Jong JH, Cameron K, Leersum RV, Rodermond HM, Franitza M, Nurnberg P, Mangiapane LR, Wang X, Clevers H, Vermeulen L, Stassi G, Medema JP. Consensus molecular subtypes of colorectal cancer are recapitulated in in vitro and in vivo models. Cell Death Differ. 2018; 25: 616-33.
- Liu H, Ren G, Wang T, Chen Y, Gong C, Bai Y, Wang B, Qi H, Shen J, Zhu L, Qian C, Lai M, Shao J. Aberrantly expressed Fra-1 by IL-6/STAT3 transactivation promotes colorectal cancer aggressiveness through epithelial-mesenchymal transition. Carcinogenesis. 2015; 36: 459-68.
- Rokavec M, Oner MG, Li H, Jackstadt R, Jiang L, Lodygin D, Kaller M, Horst D, Ziegler PK, Schwitalla S, Slotta-Huspenina J, Bader FG, Greten FR, Hermeking H. IL-6R/STAT3/miR-34a feedback loop promotes EMT-mediated colorectal cancer invasion and metastasis. J Clin Invest. 2014; 124: 1853-67.
- Zhang KW, Wang D, Cai H, Cao MQ, Zhang YY, Zhuang PY, Shen J. IL-6 plays a crucial role in epithelial-mesenchymal transition and pro-metastasis induced by sorafenib in liver cancer. Oncol Rep. 2021; 45: 1105-17.
- Yadav A, Kumar B, Datta J, Teknos TN, Kumar P. IL-6 promotes head and neck tumor metastasis by inducing epithelial-mesenchymal transition via the JAK-STAT3-SNAIL signaling pathway. Mol Cancer Res. 2011; 9: 1658-67.

Round 2
Reviewer 1 Report
Thank you for additional data and information as suggested.